# Phase-Controlled Multi-Dimensional-Structure SnS/SnS_2_/CdS Nanocomposite for Development of Solar-Driven Hydrogen Evolution Photocatalyst

**DOI:** 10.3390/ijms241813774

**Published:** 2023-09-07

**Authors:** Rak Hyun Jeong, Jae Hyeong Lee, Jin-Hyo Boo

**Affiliations:** 1Department of Electrical and Computer Engineering, Sungkyunkwan University, Suwon 440-746, Republic of Korea; jrh1015@naver.com (R.H.J.); jaehyeong@skku.edu (J.H.L.); 2Institute of Basic Science, Sungkyunkwan University, Suwon 440-746, Republic of Korea; 3Department of Chemistry, Sungkyunkwan University, Suwon 440-746, Republic of Korea

**Keywords:** 2D materials, solar-driven photocatalyst, nanocomposite, hydrogen evolution, environment pollutant removal

## Abstract

The quest for water-splitting photocatalysts to generate hydrogen as a clean energy source from two-dimensional (2D) materials has enormous implications for sustainable energy solutions. Photocatalytic water splitting, a major field of interest, is focused on the efficient production of hydrogen from renewable resources such as water using 2D materials. Tin sulfide and tin disulfide, collectively known as SnS and SnS_2_, respectively, are metal sulfide compounds that have gained attention for their photocatalytic properties. Their unique electronic structures and morphological characteristics make them promising candidates for harnessing solar energy for environmental and energy-related purposes. CdS/SnS/SnS_2_ photocatalysts with two Sn phases (II and IV) were synthesized using a solvothermal method in this study. CdS was successfully placed on a broad SnS/SnS_2_ plane after a series of characterizations. We found that it is composited in the same way as a core-shell shape. When the SnS/SnS_2_ phase ratio was dominated by SnS and the structure was composited with CdS, the degradation efficiency was optimal. This material demonstrated high photocatalytic hydrogenation efficiency as well as efficient photocatalytic removal of Cr(VI) over 120 min. Because of the broad light absorption of CdS, the specific surface area, which is the reaction site, became very large. Second, it served as a transport medium for electron transfer from the conduction band (CB) of the SnS to the CB of the SnS_2_. Because of the composite, these electrons flowed into the CB of CdS, improving the separation efficiency of the photogenerated carriers even further. This material, which was easily composited, also effectively prevented mineral corrosion, which is a major issue with CdS.

## 1. Introduction

Because of its high energy density, hydrogen gas is a promising future energy resource [1,2,3]. Furthermore, hydrogen is an environmentally friendly fuel because its combustion produces only water, whereas conventional fossil fuels emit pollutants and greenhouse gases, causing environmental destruction and global climate change [4]. As a result, the production of hydrogen gas as an environmentally friendly and renewable energy source will be a significant achievement for future environmental protection. One of the most promising approaches to producing renewable hydrogen as an energy source is solar-powered photocatalytic water splitting [5]. Significant progress has been made in the development of photocatalytic water-splitting systems, which have sparked intense interest in recent years. In recent years, there has been a lot of interest in the production of hydrogen gas using sunlight and semiconductor photocatalysts. Semiconductor-based photocatalytic technology is attracting attention due to its advantages such as eco-friendliness, stability, and the lack of need for additional energy. However, some challenges must be overcome before this technology can be used sustainably, including relatively low efficiency, stability problems against photocorrosion, and visible-light driving [6]. One important point to note is that UV light accounts for only about 5% of total energy in the solar spectrum. In contrast, visible light (wavelength 400–800 nm) accounts for more than half of solar energy, and developing long-wavelength reactive photocatalysts is critical to achieving truly efficient renewable hydrogen production [7]. As a result, the development of visible-light-driven photocatalysts is critical to the success of this technology. Many researchers have studied enhanced absorption across the entire solar spectrum, improved electron–hole separation, and accelerated charge transfer to achieve more satisfactory catalytic performance. So far, several strategies for optimizing materials have been reported, including constructing heterostructures by combining binary or multicomponent composites [8], doping [9], bandgap engineering [10], and shape modulation [11]. In the reported literature, the construction of composites has been identified as a successful method for improving photocatalytic performance [12]. Composite methods that smooth the carrier movement between interfaces by band aligning the low-solar-energy absorption band and the fast electron–hole recombination problem are being studied variously [13].

SnS is a p-type semiconductor with a broad absorption capacity in the visible-to-near-infrared region of the solar spectrum [14]. SnS is also abundant on Earth, inexpensive, and non-toxic, and has shown promise in photocatalysis and photodetectors. This material has a curved honeycomb structure similar to black phosphorus [15]. The bandgap can be adjusted based on the thickness as well as the electrical and optical properties. Because of the combination of cationic and anionic layers in an orthorhombic crystal structure of Sn and S, it exhibits anisotropy with a curved honeycomb structure and has high chemical stability [16]. The bandgap of the 2D SnS layer does not decrease monotonically but exhibits odd–even quantum confinement effects due to different energy band splits associated with different thicknesses but symmetrical fluctuations [17]. More importantly, the 2D SnS layer has significantly higher in-plane/anisotropy and mobility than BP. The SnS sheet is a very promising material as a nanoelectronic semiconductor because of its adjustable bandgap and high carrier mobility [18]. Furthermore, unlike expensive BP, SnS is advantageous for commercialization because it is inexpensive and simple to synthesize. High water stability can be a huge benefit in photocatalytic reactions that occur in humid environments [19]. Different valence states of Sn combined with Sn(SnS/SnS_2_), for example, can form heterogeneous structures that promote photolysis. For example, Chang et al. studied photodetectors with strong visible-light absorption, good photoelectric conversion efficiency, and effective control of the behavior of photogenerated carriers at low temperatures through SnS and CdS nanorod structures [20]. Furthermore, many high-performance, flexible photodetectors have been synthesized using CdS/SnS_2_ heterostructure films [21]. This is a promising method for fabricating double heterojunctions of SnS_2_ in conjunction with other semiconductor materials to increase activity and promote electron–hole separation. Pioneering research has revealed that SnS and CdS are superior candidates for forming double heterojunctions with SnS_2_ to inhibit electron–hole recombination and broaden the photoresponse range [22].

In this study, we synthesized CdS/SnS/SnS_2_ core-shell composites of three materials using solvothermal and high-energy ultrasound methods. The characteristics and advantages of these composite structures were investigated through various analysis methods. It was also found that this material exhibits excellent performance as a photocatalyst for hydrogen production through benzyl alcohol reduction and water decomposition. In this process, a mechanism was discovered that inhibits electron–hole recombination to increase the carrier lifetime, thereby improving its efficiency.

## 2. Results and Discussion

### 2.1. Material Characterization

The CdS/SnS/SnS_2_ composite was synthesized in two steps: the first was a hydrothermal reaction, and the second was a sonication step (Figure 1). The SnS sheet structure was obtained in the first step using a temperature-controlled hydrothermal reaction. CdS nanoparticles were uniformly distributed on the SnS in the following sonication step. The external appearance is similar irrespective of the temperature, but the internal composition ratio varies.

First, as shown in Figure 1, the morphologies of the CdS, SnS, and CdS/SnS/SnS_2_ samples were preferentially analyzed via FE-SEM. The CdS sample, as shown in Figure 1a, is made up of uniform nanoparticles with very small grain sizes. In addition, Figure 1b confirms that SnS is synthesized as a two-dimensional sheet. This means that the SnS obtained through this method is a layered bulk material. The surface of the synthesized nanosheet is smooth, and SnS_2_ particles are mixed on it. This laminated sheet-like material was exfoliated in NMP solution using ultrasonication. The ultrasonically bonded CdS/SnS/SnS_2_ compound (Figure 1c) was then overlaid on a sheet to reveal a rough surface. It can be created as a core-shell structure covered with CdS nanoparticles on a sheet of SnS/SnS_2_.

To better understand the microstructure and morphology of CdS/SnS/SnS_2_, TEM measurements were taken. The prepared CdS sample, as shown in Figure 2a,b, is composed of nanoparticles with a size of 10 nm or less. A lattice of CdS QDs with a diameter of several nm can be seen in Figure 2b. CdS particles have a lattice width of 0.305 nm, which corresponds to the hexagonal [101] crystal plane [23]. The surface structure of SnS/SnS_2_ can be seen in Figure 2c,d. It takes the shape of a large sheet, with lattices corresponding to SnS_2_ particles randomly distributed throughout. These SnS_2_ particles are found to be distributed in the form of small QDs. It is confirmed that the plate morphology of SnS is dominant, with only a trace of SnS_2_ present. SnS_2_ has a lattice spacing of 0.327 nm, which is attributed to the hexagonal phase’s [1 0 0] plane [24]. The microstructure of the CdS/SnS/SnS_2_ composite is shown in Figure 2e,f. It is confirmed here that CdS is distributed on the surface of the SnS/SnS_2_ structure in the form of a core-shell. The surface is covered in small CdS particles. CdS serves as the backbone for the massive structure of SnS/SnS_2_. Lattice spacing corresponding to 0.332 nm for the [0 0 2] lattice plane of CdS, 0.275 nm for the [0 4 0] lattice plane of SnS, and 0.327 nm for SnS_2_ is observed [25].

EDS analysis via TEM confirms the elemental distribution of each structure. Figure 3a depicts the CdS particles, while Figure 3b depicts the EDS data of the structure of SnS/SnS_2_. Figure 3c depicts the EDS data of the CdS/SnS/SnS_2_ compound, which is confirmed by mapping over the SnS/SnS_2_ structure. This shape improves photocatalytic performance by encouraging the movement of light-generated carriers, and it can have a larger specific surface area than the sheet-like shape of SnS.

BET measurements were taken to investigate the improved reaction site and specific surface area over the SnS (Figure 4a,b) [26]. Furthermore, the difference in specific surface area between the two materials was determined using nitrogen adsorption/desorption BET. As a result, little adsorption is observed in the SnS/SnS_2_ sheets, whereas the BET surface area value in the CdS/SnS/SnS_2_ sample is 121.8 m^2^/g. The high specific surface area of the CdS/SnS/SnS_2_ composite is beneficial for the adsorption of reactive species and reactants on the surface, which improves photocatalytic performance [27].

The crystal structure and phase of the sample were determined using XRD (Figure 5a). The XRD pattern of the synthesized SnS/SnS demonstrates high crystallinity, with diffraction peak sets of 22° [1 1 0], 26.0° [1 2 0], 31.6° [1 1 1], 32.1° [0 4 0], 39° [1 3 1], and 45.5° [1 5 0], which is due to the orthorhombic SnS (JCPDS 39-0354) [28]. In general, SnS_2_ has a hexagonal crystal structure (JCPDS 21-1231). The peaks correspond to 28.2° [1 0 0], 30.5° [0 0 2], 50.3° [1 1 0], 52.6° [1 1 1], and 58.4° [2 0 0] at the 2θ values [29]. Because of the low diffraction intensity, the diffraction peak corresponding to CdS appears as a broad peak in the XRD pattern. This is due to CdS’s relatively low crystallinity and small grain size.

Solid-state Raman spectroscopy can confirm the sample’s structure further (Figure 5b). It was not measured in Raman spectroscopy due to the low crystallinity of CdS and laser damage. As a result, the spectra of SnS/SnS_2_ and CdS/SnS/SnS_2_ composites are observed to be similar. This demonstrates that SnS/SnS_2_ was stably bound without any abnormalities during the CdS conjugation process. There are four distinct peaks at 95.9, 164.0, 192.0, and 219.5 cm^−1^ [30]. These peaks are attributed to the various Raman modes of Ag in SnS and B_3g_. Ag 40.1 cm^−1^ and Ag 219.3 cm^−1^ correspond to the layer shear mode and “NaCl”-type oscillations, respectively [31]. The other two Ag peaks, at 95.9 and 192.0 cm^−1^, occur in “shake” and “breathing” modes, respectively, whereas B_3g_ 49.1 cm^−1^ and B_3g_ 164.0 cm^−1^ are zigzag shear oscillations and “NaCl”-type oscillations [31]. Furthermore, the peak corresponding to the SnS_2_ mode is very low, near 300 cm^−1^. This is due to the fact that they exist in the form of very small quantum dots, as seen in TEM.

The samples’ Fourier transform infrared (FT-IR) spectra (Figure 5c) were measured. Similarly, the peak intensity of CdS in FT-IR is very low, and it is confirmed that bonding with SnS/SnS_2_ increased the peak of the vibration mode near 1000 cm^−1^, corresponding to the vibration of the -S bond.

The chemical composition of SnS/SnS_2_ components were identified using XPS (Figure 6). XPS analysis can provide detailed observations of a material’s surface state and composition. The XPS irradiation spectrum in Figure 6 reveals the presence of C, O, Sn, Cd, and S atoms in the prepared samples. In general, the elements C and O can be found in samples that have absorbed CO_2_ from the air [32]. Figure 6 depicts the Sn peak. The peaks at 486.7 and 495.2 eV are attributable to Sn 3d^5/2^ and 3d^3/2^ to the Sn(IV) of SnS_2_, respectively. And the two peaks at 484.7 and 493.2 eV can be attributed to Sn 3d^5/2^ and 3d^3/2^, respectively, which correspond to Sn(II) in the SnS reference data [33,34]. SnS is more dominant in Figure 6a, Figure 6b has a similar proportion, and Figure 6c is SnS_2_-dominant. As a result, the phase ratios of Sn (IV) and Sn (II) could be controlled based on the synthesis mole ratio and temperature. Sample c, which was placed on SnS_2_ in SnS sheet form, demonstrates the highest efficiency, so it was used for all the samples.

Figure 7 depicts the high-resolution XPS spectra for S 2p and Cd 3d. In Figure 7a, the XPS spectrum of Cd 3d has two peaks measured at 411.9 eV and 405.1 eV, corresponding to the binding energies of Cd 3d^3/2^ and Cd 3d^5/2^, respectively [35]. The shoulder peak closest to the left corresponds to the loss. The S 2p spectrum (Figure 7b) indicates the presence of S 2 anions, S 2p^3/2^, and S 2p^1/2^, at binding energies of 162.7 eV and 161.5 eV, respectively [36].

The photoelectrochemical properties of tin sulfide are primarily related to its light absorption properties, photogenic carrier generation, segregation, and interface transfer between the tin sulfide and the CdS [37]. UV-Vis diffuse reflection spectroscopy was used to obtain the diffuse reflectance curves of the samples. CdS/SnS/SnS_2_ absorbs more than SnS/SnS_2_ in the UV–visible region, as shown in Figure 8a. Kubelka Munc’s law was used to calculate the bandgap energies of the materials [38].
$$\frac{k}{s}=F\left(R_{\infty }\right)=\frac{{\left(1-R_{\infty }\right)}^{2}}{2R_{\infty }}$$

The bandgap value is the x-intercept of the tangent of this graph [39]. In Figure 8b–d, CdS/SnS/SnS_2_ heterojunction nanosheets absorb light in a wide range from UV to NIR. The E_g_ value of the CdS/SnS/SnS_2_ composite nanosheets is 2.98 eV, which is wider than those of SnS/SnS_2_ 1.75 eV and CdS 2.25 eV.

### 2.2. Photocatalytic Reaction Properties

We investigated whether previously synthesized SnS/SnS_2_ is a suitable material for visible-light-driven photocatalytic hydrogen generation and how it affects composite formation with CdS. The photocatalytic hydrogen generation experiment was performed using SnS/SnS_2_ and CdS, as well as SnS and SnS_2_ as controls, as shown in Figure 9. A small amount of IPA/methanol was added to the reaction solution as a sacrificial reagent to suppress the reverse reaction and increase the quantum yield. It is confirmed in Figure 9a,b that all single materials have little activity for visible-light-driven hydrogen generation. The average hydrogen generation efficiency was 35.57 μmol/h, single CdS was 1220.27 μmol/h, and when a complex was formed, the efficiency increased up to 2467.83 μmol/h. A recycling test was also performed to confirm the durability of photocorrosion, which is the most serious issue in the photocatalytic reaction of CdS materials. As a result, even though the CdS/SnS/SnS_2_ recycling test was repeated approximately six times, the photocatalytic hydrogen generation efficiency was maintained and slightly increased from the initial one.

To investigate the photocatalytic performance of the samples further, photoreduction Cr(VI) was performed [40,41]. The time-dependent reduction rates of Cr(VI) for various photocatalysts are shown in Figure 9c. A clean 15 mg/L Cr(VI) solution under illumination, as shown in the figure, indicates that no autocatalytic reduction of Cr(VI) can occur. Due to the relatively bulky SnS material’s poor light and light carrier transport properties, the majority of the conformation-dependent SnS structures demonstrate weak photocatalytic Cr(VI) reduction ability. The Cr(VI) reduction rate of SnS QDs, on the other hand, decreases to 80% of the Cr(VI) concentration at 120 min due to a change in the band whereby it becomes smaller and thinner. Furthermore, materials with two different bands are combined in CdS/SnS/SnS_2_ to prevent photocarrier recombination, and the reduction efficiency is improved due to the wide catalytic reaction site. The combination of two samples with different thicknesses and sizes, despite being made of the same material, significantly improve the light reduction performance. At 120 min, the CdS/SnS/SnS_2_ composite reduces by 100% Cr(VI), indicating excellent photoreduced Cr(VI) performance. Furthermore, the concentration of Cr(VI) drops sharply during the first 5 min before slowing down over the next few minutes. This should be caused by gradually decreasing Cr_2_O_7_ levels. In Figure 9d, a recycling experiment was carried out by continuously repeating the Cr(VI) photoreduction experiment of SnS MS. Even after 5 iterations, this demonstrates consistent performance.

There are several reasons why the photocatalytic efficiency and stability of the CdS/SnS/SnS_2_ composite are increased. Once on the advantage of the SnS/SnS_2_ mixed material is that the bonding with SnS inhibits the reduction reaction of the pure SnS_2_ material because the electrons for SnS_2_ provided via photogeneration are accepted rather than transferred to the Sn^4−^ ion of the SnS_2_ lattice [42]. Therefore, the stability can be further improved. In addition, because the Cds/SnS/SnS_2_ sample has a much larger specific surface area than standard SnS/SnS_2_, more photocatalytic reactions can take place on the surface. Because photocatalysts are studied primarily through surface chemistry, a large surface area is a highly favorable condition for photocatalytic reactions [43]. Second, the photocatalytic efficiency was increased by lowering the recombination rate of photoinduced electron–hole charge carriers in the complex system and the increasing lifetime. To compensate, the carrier lifetimes of the samples were measured using PL and TRPL (Figure 10a,b) [44,45,46]. As a result, the photoinduced charge carrier lifetime of CdS/SnS/SnS_2_ was longer than that of conventional single materials, and a 60% increase in carrier lifetime was confirmed. This follows the same pattern as the photocatalyst efficiency graph. Finally, to explain the photocatalytic reaction mechanism, a schematic diagram of the photoinduced charge carrier transport process through the TP/RP sample interface under visible-light irradiation was proposed (Figure 10c). The photocatalytic reaction mechanism is shown below. First, broad light absorption by CdS on the surface of the sample particle results in the formation of electron holes [47]. It then forms an electron by reacting with an unshared oxygen radical (e^−^). At this point, the electrons move to the adjacent CdS’s conduction band (CB) and react with oxygen to form a peroxide anion. Furthermore, they positively react with water molecules on the particle surface to form OH radicals, which then revert to their original state [48]. The composite samples outperformed the CdS or SnS single samples in terms of photocatalytic hydrogen evolution. This improvement in photocatalytic activity performance can be explained as follows. First, because the VB edge positions of SnS/SnS_2_ are higher than those of CdS, holes can easily move through the interface to the VB of the SnS. The migrated carriers can reduce the probability of recombination, effectively separating the charge carriers and improving photocatalytic efficiency [49]. The composite structure’s interface is also important in the separation efficiency of photoinduced electrons and holes [50]. Furthermore, in the case of CdS, fast photocorrosion is a disadvantage, but SnS/SnS_2_ acts as an acceptor, improving stability.

## 3. Materials and Methods

### 3.1. Materials

CdS nanoparticles, Dimethylformamide (DMF), Thioacetamide(C_2_H_5_NS), Tin(II) chloride, Tin(IV) chloride, Potassium dichromate(VI), *N*-methyl-2-pyrrolidone (NMP), and all other chemicals were acquired from Sigma-Aldrich (St. Louis, MO, USA) and used without further purification. Distilled water was purified using a tertiary filter, and ethyl alcohol (98%) and acetone (99%) were used as solvents.

### 3.2. Preparation of CdS/SnS/SnS_2_ Nanocomposite

SnS/SnS_2_ nanosheets were synthesized using the solvothermal method. In 38 mL of DMF, SnCl_2_, SnCl_4_, and Thioacetamide were each adjusted in molar ratio and mixed under vacuum for 1 h. Then, we put the solution in a 50 mL Teflon autoclave to react at 200 °C for 12 h. After cooling naturally, it was washed three times with ethanol and dried via centrifugation. The exfoliation process was performed via high-energy sonication of N-methyl-2-pyrrolidone (NMP) solution while cooling the bath.

We dispersed the synthesized SnS/SnS_2_ nanosheet and CdS nanoparticles in a mixed solution of ethanol and distilled water. After this, high-energy ultra horn sonication was performed in an ice bath for 5 h. Finally, high-speed centrifugation was performed and the sample was washed with ethanol 3 times.

### 3.3. Photocatalytic Hydrogen Evolution Measurements

The photocatalytic hydrogen evolution test was used to perform hydrogen evolution photocatalytic activation in a reaction vessel using a 250 W Xe lamp equipped with a 400 nm cut-off filter. Distilled water and methanol/IPA were mixed in a photoreactor in a volume ratio of 5:1:1, and 0.2 g of photocatalyst powder was added and stirred so as to be uniformly dispersed. Before irradiation, the reaction solution was evacuated several times to remove air, and the reaction temperature was always maintained at 279 K. To determine the amount of H_2_ produced, gas chromatography (YoungIn Chromass GC6500 system, PDD detector, Anyang, Republic of Korea) was performed using high-purity He as a carrier gas.

### 3.4. Photocatalytic Organic Pollutant Degradation

The photocatalytic reduction of Cr(VI) was carried out using 15 mg/L of a 50 mL K_2_Cr_2_O_7_ solution and 10 mg photocatalyst, as well as 20 vol% triethanolamine (TEOA) as a sacrificial agent. To achieve adsorption/desorption equilibrium, the K_2_Cr_2_O_7_ solution containing the photocatalyst was stirred in the dark for 30 min before light irradiation, and the temperature of the Cr(VI) photoreduction reaction was also maintained to avoid thermal effects. A solar simulator (Datech’s Model-DXP300, Seoul, Republic of Korea) with a UV cut filter (output wavelength: 400–1000 nm) was used as the light source. A total of 4 mL of the reaction solution was obtained from the reactor at 0 °C at 5 min intervals and centrifuged at 10,000 rpm for 5 min to collect the supernatant. The Cr(VI) concentration was determined as follows. Finally, UV-Vis spectroscopy was used to examine the magenta complex, which has a distinct absorption peak at 540 nm. Following this, a recycling test was carried out as follows. First, the sample was washed and centrifuged with distilled water after the photocatalyst was measured. To remove the dye from the sample surface, the procedure was repeated several times. For each photocatalyst reaction time, five recycle tests were performed.

### 3.5. Characterization

Field emission scanning electron microscopy (FE-SEM, Model JSM-7100 F, JEOL, Tokyo, Japan) was used to evaluate the surface morphologies of all samples. Cs-corrected transmission electron microscopy (Cs correct TEM, Model JEM ARM 200F, JEOL, Tokyo, Japan) at an accelerating voltage of 80–200 kV was used to examine the lattice structures and exfoliation status of all samples. This unit included an EM unit, a high-angle annular dark-field detector, an FLC unit, an ultra-scan charge-coupled detector (CCD) camera unit, and an energy-dispersive spectrometry (EDS) unit. X-ray diffraction (XRD, D/Max Ultima III Rigaku Corporation, Tokyo, Japan) was used to determine the crystallinity of the material. UV-Vis absorption spectroscopy was used to monitor the catalytic decomposition of Cr(VI) (UV-3600 Plus UV-Vis-NIR Spectrophotometer, SHIMADZU, Kyoto, Japan). In addition, UV-Vis absorption techniques were used to assess photocatalytic stability and catalyst capacity. A laser with a 532 nm wavelength was linearly polarized from the confocal Raman spectroscopy measurements (NTEGRA Spectra, NT-MDT Co., Zelenograd, Russia). The objective lens used in all experiments had 0.7 NA and a magnification of 100× (Mitutoyo, Japan). A CCD (Andor, Abingdon-on-Thames, UK) cooled to 75 °C and a spectrometer with a grating of 1800 grooves/mm blazed at 500 nm were used to obtain Raman scattering signals.

## 4. Conclusions

The solvothermal method was used to synthesize a composite by controlling the phase of Sn and combining it with CdS. Sonication was used to separate the two materials for nanocomposite synthesis. Various spectroscopy methods were used to examine this material. In a solar simulator equipped with a UV cut-off filter, our composite catalyst demonstrated excellent photocatalytic performance for high-concentration water -splitting hydrogen production and Cr(VI) reduction (wavelength 400–1000 nm). The efficiency of hydrogen production increased to 2467.83 umol/h, which is far superior to other reported photocatalysts. Furthermore, even after up to six cycles of the photocatalytic reaction, no reduction in efficiency or material deterioration was observed. This increased photoactivity and stability are attributed to the two materials’ large specific surface area, anti-recombination, and fast conductivity effects.

We presented a method for simultaneously solving the problem of CdS stability and fast and dominant electron–hole recombination in this study. When we applied two-dimensional SnS to the photocatalysts, we discovered a lot of potential and possibilities. Building a database of various materials will have a positive impact on the design of photocatalysts for other purposes when designing photocatalyst materials. Furthermore, the oxidation control of BP via plasma treatment is expected to be applicable to other materials. We believe that this research opens new possibilities for the various two-dimensional-material photocatalysts that are currently being investigated. Furthermore, the development of a hybrid photocatalyst material that can decompose both air and water pollutants at the same time could be a fundamental solution to the environmental pollution problem.

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
