# Peer review of "Phase-Controlled Multi-Dimensional-Structure SnS/SnS2/CdS Nanocomposite for Development of Solar-Driven Hydrogen Evolution Photocatalyst"

_ijms, 2023, doi:10.3390/ijms241813774_

Round 1

Reviewer 1 Report

The article may be accepted after addressing the following comments/correction:
1. For comparison purposes, please provide the SEM image of 1 (a) and 1(c) with the same magnification with better quality.  2. This manuscript does not contain much error analysis of the degradation of the Cr(VI)  which is highly needed for readability purposes. The error bars of the degradation of the Cr(VI) is needed so that the reader will have an idea of the reproducibility of the data.  3. What about the morphology and crystal structure of the photocatalyst after the recycling? 4. What about the dark adsorption of  Cr(VI) over the synthesized catalyst? 5. The authors need to cite other photocatalysts, for example, the degradation of MO and MB by TiS3 nanostructures (https://doi.org/10.1016/j.catcom.2021.106381); bifunctional 2D/2D heterojunction photocatalyst for cooperative H2 production and alcohol conversion (https://doi.org/10.1016/j.apcatb.2023.122725).

Author Response

To

The Editor -in -Chief,

International journal of molecular sciences

Dear Sir,

Here with I am submitting the revised version of the article entitled “Phase controlled multi-dimensional structure SnS/SnS2/CdS nanocomposite for solar-driven hydrogen evolution photocatalyst by Rak Hyun Jeong and Jin-Hyo Boo modified according to the suggestions of the referees.

Modifications made in the present manuscript are listed below.

  • Comments of the Reviewers have been answered (Next page: Replies to Reviewers comments) and incorporated in the manuscript.
  • Two references were added. (Ref. 40 and 41)
  • Correction of typo and English errors.
  • Figures are modified. (Figure1 a, Figure6, Figure 9)

With the above modifications and other corrections, the manuscript has been improved to a great extent. We believe that the present version will fulfill the expectations of the referees and meet the quality standards of the International journal of molecular sciences. Thank you so much for your interest and valuable suggestions to improve the manuscript. Please do not hesitate to ask questions if you do not have enough answers or if you have any additional questions. We will do our best to respond that as a top priority.

Kindly consider this manuscript for the publication in your esteemed journal.

Yours sincerely,

Jin-Hyo Boo.

Reply to the Referees comments

Referee 1

1. For comparison purposes, please provide the SEM image of 1 (a) and 1(c) with the same magnification with better quality.

Reply: We sincerely appreciate the comment of the reviewer. Applying the reviewer's comments, Figure1 (a) was replaced to the high-resolution SEM image with same magnification. We have faithfully fulfilled the instructions given in the comments.

2. This manuscript does not contain much error analysis of the degradation of the Cr(VI) which is highly needed for readability purposes. The error bars of the degradation of the Cr(VI) is needed so that the reader will have an idea of the reproducibility of the data.

Reply: We sincerely appreciate the comment of the reviewer. Applying the reviewer's comments, error bars were added to the data on Cr(VI) degradation. We have faithfully fulfilled the instructions given in the comments.

3. What about the morphology and crystal structure of the photocatalyst after the recycling?

Reply: As a result of measuring the condition of the sample after repeating the photocatalysis process 5 times with the same measurement method, no change such as deterioration or photocorrosion was observed with the 5 repetitions of the experiment.

Since SnS and SnS2 are very stable base materials, the 3 materials of heterostructures developed in this study can expect longer catalyst durability.

4.What about the dark adsorption of Cr(VI) over the synthesized catalyst?

Reply: Thanks to kind instruction. In the experiment of Cr(VI) reduction photocatalyst experiment, light is irradiated after sufficiently adsorbed in a dark room and stirred until equilibrium (30 minutes). In this process, the amount of adsorption can be found. - As shown in Figure 9(c), the adsorption amount is about 2~3% for 30 minutes in the dark state before light irradiation.

 5. The authors need to cite other photocatalysts, for example, the degradation of MO and MB by TiS3 nanostructures (https://doi.org/10.1016/j.catcom.2021.106381); bifunctional 2D/2D heterojunction photocatalyst for cooperative H2 production and alcohol conversion

 Reply: Thanks to kind instruction. We add two references (Ref. 40,41) to the content. Thanks to kind instruction. The revised section will be highlighted according to the opinion of the reviewer.

Reviewer 2 Report

In the manuscript entitled “Phase controlled multi-dimensional structure SnS/SnS2/CdS nanocomposite for solar-driven hydrogen evolution photocatalyst” authors synthesized CdS/SnS/SnS2 core-shell composites of three materials using solvothermal and high-energy ultrasound methods.

The manuscript is well articulated, the arguments are presented in a manner consistent with the hypotheses formulated, and the literature review is satisfactory and up-to-date.

Despite the manuscript is well written and it is potentiality of being shared with the scientific community, I believe that it would benefit from a minor revision.

Main concern:

ABSTRACT The authors should start with a short intro that better highlights their work and end up with a paragraph that include results. It should include Background, Aim or purpose of research, Method used, Findings/results, Conclusion.

RESULT AND DISCUSSION: Authors should separate Result and Discussion in two different paragraphs. The discussion should start with a first paragraph describing the main aims and then the main results.

Author Response

August 30, 2023

To

The Editor -in -Chief,

International journal of molecular sciences

Dear Sir,

Here with I am submitting the revised version of the article entitled “Phase controlled multi-dimensional structure SnS/SnS2/CdS nanocomposite for solar-driven hydrogen evolution photocatalyst by Rak Hyun Jeong and Jin-Hyo Boo modified according to the suggestions of the referees.

Modifications made in the present manuscript are listed below.

  • Comments of the Reviewers have been answered (Next page: Replies to Reviewers comments) and incorporated in the manuscript.
  • Two references were added. (Ref. 40 and 41)
  • Correction of typo and English errors.
  • Figures are modified. (Figure1 a, Figure6, Figure 9)

With the above modifications and other corrections, the manuscript has been improved to a great extent. We believe that the present version will fulfill the expectations of the referees and meet the quality standards of the International journal of molecular sciences. Thank you so much for your interest and valuable suggestions to improve the manuscript. Please do not hesitate to ask questions if you do not have enough answers or if you have any additional questions. We will do our best to respond that as a top priority.

Kindly consider this manuscript for the publication in your esteemed journal.

Yours sincerely,

Jin-Hyo Boo.

Reviewer 2

In the manuscript entitled “Phase controlled multi-dimensional structure SnS/SnS2/CdS nanocomposite for solar-driven hydrogen evolution photocatalyst” authors synthesized CdS/SnS/SnS2 core-shell composites of three materials using solvothermal and high-energy ultrasound methods.

The manuscript is well articulated, the arguments are presented in a manner consistent with the hypotheses formulated, and the literature review is satisfactory and up-to-date.

Despite the manuscript is well written and it is potentiality of being shared with the scientific community, I believe that it would benefit from a minor revision.

Main concern:

ABSTRACT The authors should start with a short intro that better highlights their work and end up with a paragraph that include results. It should include Background, Aim or purpose of research, Method used, Findings/results, Conclusion.

RESULT AND DISCUSSION: Authors should separate Result and Discussion in two different paragraphs. The discussion should start with a first paragraph describing the main aims and then the main results.

Reply: We are genuinely pleased and thankful for the positive opinion of the reviewer.

We modified the part in the abstract section for your comments, and also corrected the part of result and discussion.

Reviewer 3 Report

I found this paper very interesting. However, the authors should bring more care to the presentation.

- Lines 71-74: For example...were also studied [20]. The authors should rewrite these lines in an understandable way.

- line 110: hours. Finally (the space was missing)

line 117: In following are described...

- line 2153: Figure 6. : the synthesis molar ratios are not shown

- line 264: [33,34].

- line 523: Ref. 38 is not complete

See the corrections in the previous Comments.

Author Response

August 30, 2023

To

The Editor -in -Chief,

International journal of molecular sciences

Dear Sir,

Here with I am submitting the revised version of the article entitled “Phase controlled multi-dimensional structure SnS/SnS2/CdS nanocomposite for solar-driven hydrogen evolution photocatalyst by Rak Hyun Jeong and Jin-Hyo Boo modified according to the suggestions of the referees.

Modifications made in the present manuscript are listed below.

  • Comments of the Reviewers have been answered (Next page: Replies to Reviewers comments) and incorporated in the manuscript.
  • Two references were added. (Ref. 40 and 41)
  • Correction of typo and English errors.
  • Figures are modified. (Figure1 a, Figure6, Figure 9)

With the above modifications and other corrections, the manuscript has been improved to a great extent. We believe that the present version will fulfill the expectations of the referees and meet the quality standards of the International journal of molecular sciences. Thank you so much for your interest and valuable suggestions to improve the manuscript. Please do not hesitate to ask questions if you do not have enough answers or if you have any additional questions. We will do our best to respond that as a top priority.

Kindly consider this manuscript for the publication in your esteemed journal.

Yours sincerely,

Jin-Hyo Boo.

Reviewer 3

I found this paper very interesting. However, the authors should bring more care to the presentation.

- Lines 71-74: For example...were also studied [20]. The authors should rewrite these lines in an understandable way.

- line 110: hours. Finally (the space was missing)

line 117: In following are described...

- line 2153: Figure 6. : the synthesis molar ratios are not shown

- line 264: [33,34].

- line 523: Ref. 38 is not complete

.

Reply: First of all, we appreciate the positive feedback of the reviewer.

Thanks to kind instruction. According to the Reviewer's comments, we've made all applicable changes. In addition, we found and corrected all the typo.

- Lines 71-74: Rewritten.

- Line 110: Corrected.

Line 117: Removed unnecessary sentences.

- Line 2153: Added.

- Line 264: Corrected.

- Line 523: Added.

Round 2

Reviewer 1 Report

All comments/corrections have been addressed. The quality of the manuscript has improved. Article may be accepted now.